# Adenosine 5′-Monophosphate-to-Threonine Ratio Promotes Abdominal Aortic Aneurysms via Up-Regulation of HLA-DR on Natural Killer Cells: A Bidirectional Mendelian Randomized Analysis

**DOI:** 10.3390/biomedicines12061179

**Published:** 2024-05-25

**Authors:** Fei Teng, Youyin Tang, Zhangyu Lu, Zheyu Chen, Qiang Guo

**Affiliations:** 1Division of Liver Surgery, Department of General Surgery, West China Hospital of Sichuan University, No. 37 GuoXue Alley, Chengdu 610041, China; a565073825@outlook.com; 2Division of Vascular Surgery, Department of General Surgery, West China Hospital of Sichuan University, No. 37 GuoXue Alley, Chengdu 610041, China; story_tang@foxmail.com; 3West China School of Medicine, Sichuan University, No. 17 South Renming Road, Chengdu 610094, China; 2022181620083@stu.scu.edu.cn

**Keywords:** blood immune cell, metabolites, abdominal aortic aneurysm, mediation analysis, Mendelian randomization

## Abstract

**Objective**: Immune–metabolic interactions may have causal and therapeutic impacts on abdominal aortic aneurysms (AAAs). However, due to the lack of research on the relationship between immune–metabolic interactions and AAAs, further exploration of the mechanism faces challenges. **Methods**: A two-sample, two-step mediation analysis with Mendelian randomization (MR) based on genome-wide association studies (GWASs) was performed to determine the causal associations among blood immune cell signatures, metabolites, and AAAs. The stability, heterogeneity, and pleiotropy of the results were verified using a multivariate sensitivity analysis. **Results**: After multiple two-sample MRs using the AAA data from two large-scale GWAS databases, we determined that the human leukocyte antigen-DR (HLA-DR) levels on HLA-DR + natural killer (NK) cells (HLA-DR/NK) were associated with the causal effect of an AAA, with consistent results in the two databases (FinnGen: odds ratio (OR) = 1.054, 95% confidence interval (CI): 1.003–1.067, *p*-value = 0.036; UK Biobank: OR = 1.149, 95% CI: 1.046–1.261, *p*-value = 0.004). The metabolites associated with the risk of developing an AAA were enriched to find a specific metabolic model. We also found that the ratio of adenosine 5′-monophosphate (AMP) to threonine could act as a potential mediator between the HLA/NK and an AAA, with a direct effect (beta effect = 0.0496) and an indirect effect (beta effect = 0.0029). The mediation proportion was 5.56%. **Conclusions**: Our study found that an up-regulation of HLA-DR on HLA-DR/NK cells can increase the risk of an AAA via improvements in the AMP-to-threonine ratio, thus providing a potential new biomarker for the prediction and treatment of AAAs.

## 1. Introduction

Abdominal aortic aneurysm (AAA) is a cardiovascular disease characterized by the dilation of the abdominal aorta by more than 50% of its original size [1,2]. Immune-mediated inflammation and destruction of the aortic wall are involved in the onset and development of the disease [3]. The incidence of AAA is reported to be approximately 0.6‰ in America, and approximately 200,000 new AAA cases occur each year [4]. Unfortunately, due to aggravation by population aging, the prevalence of AAA will certainly increase. Smoking, old age, hypertension, atherosclerosis, hypercholesterolemia, and male gender are recognized risk factors for AAA [5]. Due to the nature of the asymptomatic characteristics in most AAA patients and the extremely high mortality rate when an AAA ruptures, the early diagnosis and prevention of AAAs face significant challenges.

With the increase in studies on the pathogenesis of AAA this year, researchers have found that in addition to smoking, COPD, atherosclerosis, and genetic susceptibility are the main risk factors for the development of AAA and that the immune system and metabonomics play crucial roles in the occurrence and development of AAA [6,7,8,9]. In addition, the abnormal activation of immune cells and the degenerative changes in elastic fibers and collagen fibers are some of the main factors leading to the formation of an AAA [4,10]. Previous studies have also shown that some immune-related biomarkers can serve as indicators for predicting the progression of AAA, such as CX3CR1 and HBB [11]. Metabolomics can provide a deeper understanding of the biological mechanisms of diseases by identifying modified metabolites or metabolic pathways, providing new insights into the prevention and developmental mechanisms of AAA. In a recent metabolomics study related to AAA, it was found that alpha-ketoglutarate may affect the formation of an AAA by affecting the mitochondrial tricarboxylic acid cycle. Although researchers have reported that specific types of metabolites can affect immune function [12,13,14], currently, few of them have studied the combination of both immune cells and metabolomics to determine the developmental mechanism of AAA [12,13,14]. In addition, limited by potential confounding factors, the observational nature of most studies, and the small sample sizes, the findings concerning the relationships among immune cells, metabolites, and AAA are still unclear.

Mendelian randomization (MR) is an analytical method based on Mendel’s law of independent assortment that provides reliable causal evidence about the relationship between an exposure and an outcome through genetic variation. The core aim of MR is to evaluate the causal impact of genetic proxy exposure on outcomes by selecting single nucleotide polymorphisms (SNPs) associated with exposure as instrumental variables (IVs) [15,16]. The IV substitution method can provide MR with a similar effect to randomized controlled trials (RCTs), as SNPs are randomly assigned to offspring during pregnancy. Thus, confounding factors can largely be avoided [17]. Moreover, the relationships among immune cells, metabolites, and AAA can be explored using a mediation analysis with MR. 

Since the relationships among immune cells, metabolites, and AAA are unclear, we hypothesized that the effect of immune cell regulation on AAA is mediated by some unknown metabolites. Thus, we performed a two-sample, two-step MR to investigate the causal role of 1400 metabolites in the effect of immune cells on AAA. We finally identified the causal effects of immune cell signatures, metabolites, and AAA, which could provide a new method for exploring the developmental mechanism of AAA.

## 2. Materials and Methods

### 2.1. Overall Study Design

The present study was approved by the Ethics Committee of the West China Hospital of Sichuan University. Patient informed consent was waived because no individual information is disclosed in this study. 

This study was designed to determine the causal effect between peripheral blood immune cells and AAA and to find their potential mechanisms via a mediation analysis with MR. The MR followed 3 basic assumptions: (1) the IVs were in valid relationships with exposure; (2) the potential confounders could not be affected by the IVs; and (3) the influence of the IVs on the outcome could only occur via exposure [18]. 

Our study was designed with these 3 assumptions in mind using SNPs from large-scale GWAS databases and followed the Strengthening the Reporting of Observational Studies in Epidemiology using Mendelian Randomization (STROBE-MR) guideline. In the filtration stage, we conducted a two-sample MR between 731 types of immune cells and AAA separately. The immune cells identified as being statistically significant in both the UK Biobank and FinnGen databases were then included in a reverse MR analysis to detect if any reverse causality existed. Then, we conducted a two-sample MR between a key cell that was fully tested on 1400 types of peripheral blood metabolites and those 1400 types of peripheral blood metabolites tested on AAA to find any potential mediators between immune cells and AAA. The overall design is shown in Figure 1.

### 2.2. Exposure and Outcome Data Sources

The 731 types of peripheral blood immune cell phenotypes were divided into 7 groups, extracted from more than 22 million SNPs tested in 3757 individuals of European ancestry. All of the GWAS phenotype summary-level data can be obtained from the GWAS Catalog website (https://www.ebi.ac.uk/gwas, accessed on 1 February 2024) [19]. The 731 immune cell phenotypes included the absolute cell counts, the relative cell counts, the surface biomarker levels on immune cells, and their morphological parameters. All of the data expressions were inputted with a Sardinian sequence-based reference panel and were adjusted for covariates like gender and age. The participants’ immune cell phenotypes had no overlap with the UK Biobank and FinnGen participant populations.

The 1400 types of peripheral blood serum metabolic phenotypes were obtained from the Canadian Longitudinal Study of Aging (CLSA) project, and a portion of the metabolic phenotypes were derived from the studies by Chen et al. [20], which included 8192 Canadians with more than 15 million summary-level SNP data inputted using a genome-wide genotyping array. The metabolic phenotypes involved the ratios between the basic chemical compounds, undiscovered chemical compounds, and the 309 types of compounds. All of the metabolite levels were quantified using the Metabolon HD4 platform and under strict quality control with systemic artifacts, misassignments, and background removal to increase the reliability.

The GWAS data for AAA were archived in the UK Biobank consortium, which was published in the GWAS Catalog (ncase = 556, ncontrols = 455,792) by [21], and in the FinnGen consortium (ncase = 3869, ncontrols = 381,997) [22]. Over 10 million SNPs were detected, and after quality control concerning the participants of the two consortiums, all the SNP effect data were at the summary level, with no personal genomic information leakage. Considering the low number of cases of AAA in the UK Biobank database, the exploration of risk factors for AAA was carried out in the FinnGen database, and the UK Biobank database was defined as an external replication.

### 2.3. Instrument Variables

The SNPs used as IVs were extracted based on a significant (*p*-value < 5 × 10^−5^) association with exposure in both the two-sample MR and multivariable MR (MVMR). The R^2^ threshold for SNP linkage disequilibrium (LD) was <0.001 within a distance of 10,000 kb. All of the LD calculations were conducted using the PLINK software (version v1.90), with the reference panel of choice being based on 1000 individuals of European ancestry from the Genomes project. The F-statistic for every SNP used as an IV, calculated as F = R^2^ × [(N − 1 − k)/k] × (1 − R^2^), should be above 10 to avoid weak instrument bias. During the calculation, R^2^ was considered the total variance of the chosen SNPs, n referred to the sample size, and k equaled the number of included SNPs. After data processing, the SNPs used as IVs for peripheral blood immune cells, peripheral blood metabolites, and AAA were included in the bidirectional MR. Finally, we excluded any IVs that were relevant confounders, such as smoking, drinking, etc., using the PhenoScanner website.

### 2.4. Mendelian Randomization Analysis

In the two-sample MR (both were bidirectional MRs), the inverse variance-weighted (IVW) method was used to preliminarily estimate whether statistical significance (*p*-value < 0.05) existed in the causal relationship between exposure and outcome. The MR–Egger method, weighted median, simple mode, and weighted mode were used in the analysis of the causal effects, in addition to MR. The results are presented in the Appendix A. Only when more than 3 SNPs acted as IVs were the results of the IVW method stable. The causal effects were calculated using an odds ratio (OR) and a 95% confidence interval (CI) to describe the degree of the effect. The heterogeneity was estimated using Cochran’s Q test result. Heterogeneity could lead to an unstable result. A random effects model was used in the MR in our study to correct for bias. The pleiotropy residual sum was measured using the MR–PRESSO and MR–Egger methods. The occurrence of pleiotropy was considered nonsense. The horizontal pleiotropy was determined based on whether statistical significance was identified in the intercept term of the MR–Egger method. A *p*-value of >0.05 identified in both the MR–Egger and MR–PRESSO methods was considered to be at no risk of pleiotropy. 

The mediation effect in the MR was evaluated using the following formula: Direct effect (beta) = Total effect (beta) − Indirect effect (beta). The mediation proportion was calculated using the indirect effect (beta) divided by the total effect (beta). The total effect was evaluated in a direct two-sample MR between exposure and outcome. The indirect effect was evaluated using the following formula: Indirect effect = beta-1 × beta-2 (beta-1 is the effect of exposure to mediating variables; beta-2 is the effect of the mediators on the outcome). The Steiger test was performed to assess any confounding bias in the exposure and mediating factors between the extracted IVs to ensure the robustness of the mediation results.

### 2.5. Metabolite Enrichment

Peripheral blood metabolic phenotypes that were found to have a significant causal effect on the risk of an AAA were extracted into an increasing AAA risk factor group and a protective group. Additionally, the metabolic phenotype ratios were removed from the enrichment. We conducted a separate enrichment analysis of the metabolites in the AAA risk and protective groups using a website (https://metaboanalyst.ca/MetaboAnalyst/home.xhtml, accessed on 24 February 2024) to determine the key metabolic pathways and patterns of AAA formation.

### 2.6. Statistical Analysis

Statistical analysis was performed using R software (version 4.3.1). The IVW method, MR–Egger, weighted median, simple mode, and weighted mode were applied using the “TwoSampleMR” and “Mendelian Randomization” packages. The MR–PRESSO package was used to find any potential pleiotropy. The “ggplot2” package was used for the funnel and forest plots. The effect sizes in this study were presented as ORs and 95% CIs. The significance threshold was set to a *p*-value of <0.05. The results between exposure and outcome at both the exploration stage and the replication stage were significant but regarded as ascertain.

## 3. Results

### 3.1. Key Immune Cell Phenotypes with Causal Relationships with AAA

We selected the IVs for the 731 immune cell phenotypes under the criterion of being relevant at a *p*-value < 5 × 10^−5^ with no obvious LD (R^2^ < 0.001, window = 10,000 kb). The immune cell phenotypes with IVs less than three SNPs were then removed in the next analysis. After filtration, we obtained 728 immune cell phenotypes for the next analysis. Any significant or other selected data for each phenotype are presented in the Appendix A. In the two-sample MR between the immune cell phenotypes and AAA from the FinnGen data, 46 phenotypes were found to have causal relationships with the risk of AAA. However, one of them (ebi-a-GCST90001803) was identified with horizontal pleiotropy. The full list shows that 27 types of peripheral blood immune-related protein markers on immune cell surfaces, 11 types of absolute lymphocyte counts, 5 types of absolute leukocyte counts, and 1 type of myeloid white cell count (CD33^−^/HLA/DR^+^ absolute count) may affect the risk of AAA. Similarly, the two-sample MR between the 728 immune cell phenotypes and the UK Biobank database showed that 35 immune cell phenotypes had causal relationships with AAA, and no pleiotropy effect was identified in these 35 MR analyses. Among the 35 types of immune cell phenotypes, 23 types of protein on immune cell surfaces, 3 types of absolute leukocyte counts, 8 types of lymphocyte counts, and the ratio of CD4^+^CD16^+^ monocytes to total monocytes were correlated with AAA. We compared the 45 and 35 types of key potential phenotypes in the FinnGen database and the UK Biobank database, respectively, to check the robustness of the results. The 22 SNPs and 25 SNPs that acted as IVs for the HLA-DR/NK levels were selected from the FinnGen database and the UK Biobank database, respectively, for the MR analysis. The results were consistent with those found in the FinnGen database (OR = 1.054, 95% CI: 1.003–1.067, *p*-value = 0.036) and in the UK biobank database (OR = 1.149, 95% CI: 1.046–1.261, *p*-value = 0.004). 

To avoid any reverse bias, we also conducted a reverse MR between the HLA-DR/NK levels and AAA from the two databases (HLA-DR/NK served as the exposure, and AAA served as the outcome from each database). The clumping threshold of AAA was set at a *p*-value of< 5 × 10^−6^, with r^2^ < 0.001, windows threshold = 10,000 kb, and F statistics > 10. Seventy-four SNPs were selected as IVs for AAA from the FinnGen database, and fifty-seven SNPs were selected as the AAA data from the UK Biobank database. No significant reverse effect was found in either of the two databases (FinnGen database: *p*-value = 0.89; UK Biobank database: *p*-value = 0.13) (Figure 2). The main results of the reverse MR analysis and single SNP effect are presented in the Appendix A.

### 3.2. Key Metabolic Phenotypes of AAA Risk

A total of 1400 types of peripheral blood metabolic phenotypes from individuals of European ancestry were selected. The SNP selection methods for the IVs were used on the immune cell phenotypes. Similarly, IVs with less than three SNPs were removed. After filtration, 1352 types of metabolic phenotypes were considered as types of exposure for the next two-sample MR. Considering the low number of cases of AAA in the UK Biobank database, we only conducted a two-sample MR on the AAA data from the FinnGen database for the outcome. The results of the MR showed that 66 types of metabolic phenotypes had causal relationships with AAA. Among the 66 types of metabolites, 29 types showed protective effects and 37 types showed opposite effects. We derived the primary metabolites’ HMDB ID and enriched them using a metabolic pathway mapping comparison with the published results for 29 protective metabolic phenotypes and 37 risky metabolic phenotypes, with the phenotypes of the ratios and any unnamed IDs in the HMDB removed.

We found that the protective metabolic phenotypes were mainly enriched for classical antibody-mediated complement activity (*p*-value < 0.01). This was correlated with a complement cascade and the initial triggering of the complement, which may represent immune regulation. The risky metabolic phenotypes mainly focused on a defective SLC6A19, which causes Hartnup disorder, and the cyclosporine A-mediated pathways (both *p*-value < 0.01). The risky metabolic phenotypes and the blood metabolic profiles of various diseases were compared. We found that the risky AAA metabolite model was correlated with branched-chain keto acid dehydrogenase deficiency, dihydrolipoamide dehydrogenase deficiency, and phenylketonuria (*p*-value < 0.001) (Figure 3). 

### 3.3. Causal Relationships between HLA on NK Cell Levels and Risky Metabolic Phenotypes of AAA


The peripheral blood metabolic phenotypes with significant causal relationships with AAA were archived for the discovery of a potential metabolic mediator effect between HLA-DR/NK and AAA. The HLA-DR/NK acted as a method of exposure with the same criterion for SNP selection: 26 SNPs were selected for the next two-sample MR. The results showed that the AMP-to-threonine ratio had a causal effect with AAA (OR = 1.029, 95% CI: 1.004–1.054, *p*-value = 0.021), which may implicate a model of energy deficiency, with no pleiotropy found and no reverse bias discovered.

### 3.4. Mediation Effect Estimation

After the two-sample MR analysis and reverse analysis, we found that the AMP-to-threonine ratio had an essential role in mediating the HLA-DR/NK cells and regulating the risk of AAA. The indirect effect of HLA-DR/NK on AAA was 0.0029, the direct effect of HLA-DR/NK to AAA was 0.0496, and the mediation proportion was 5.56% (Figure 4). The evaluation of potential reverse causal bias in the mediating effect is shown in Figure 4, and the results show that no significant reverse causal effect is present (Figure 4). 

### 3.5. Heterogeneity and Sensitivity Analysis

A funnel plot and a Cochran’s Q test were performed in our study for each two-sample MR and reverse MR. All the Cochran’s Q test results and funnel plots are presented in the Appendix A. The MR–PRESSO *p*-value and MR–Egger intercept *p*-value were used to describe any potential pleiotropy in the MR. The single SNP acting as an IV was tested via leave-one-out validation to find outliers. All the results related to pleiotropy are presented in the Appendix A. All the results were obtained following SNP duplication with obvious heterogeneity and a horizontal pleiotropy risk. With all of the pleiotropy risk removed, we obtained final results that are consistent and robust.

## 4. Discussion

This is the first time that a bidirectional two-sample mediation analysis using MR has been conducted based on a large amount of public genetic data and the possible causal relationships between multiple immune phenotypes, metabolites, and AAA have been investigated after excluding the presence of pleiotropy as much as possible. In this study, we analyzed the causal relationships among 731 immune cell signatures, 1400 metabolites, and AAA and found that the up-regulation of HLA-DR on HLA-DR/NK cells can increase the risk of AAA via improvements in the AMP-to-threonine ratio. 

In our study, we found that the increased expression of HLA-DR on HLA-DR/NK cells has a unidirectional promoting impact on AAA. This is consistent with the results of previous studies. HLA-DR is an immune cell surface receptor encoded by the human leukocyte antigen gene and plays an important role in the formation of AAA [23]. HLA-DR down-regulation is associated with hematologic malignancies, which could help with escaping from host immune surveillance [24]. However, the up-regulation of HLA-DR is related to the occurrence and development of AAA, but a prospective observational study also pointed out an interesting phenomenon: a decrease in the percentage of HLA-DR monocytes in patients who died after AAA surgery may indicate an increased risk of death, which may suggest that the expression of HLA-DR in different pathological stages and immune cells can lead to complex pathological and physiological outcomes [25]. In a cohort of AAA patients from Mexico, other researchers found that the frequencies of HLA-DRB1*01 and HLA-DRB1*16 in AAA patients were significantly elevated compared with those in healthy people [26]. In addition, in an in vitro experiment, Dragana Vucevic and her colleagues found that there was a higher percentage of HLA-DR+ immune cells in AAA tissue than in plasma [27]. As for NK cells, previous studies revealed that NK cells actively participated in the formation of atherosclerosis in in vivo mouse experiments [28]. Moreover, immuno-histochemistry and a microarray analysis of human AAA tissue have found an elevated presence of NK cells in pathological AAA [4,29].

The up-regulation of HLA-DR was also associated with several autoimmune diseases, such as rheumatoid arthritis [30], endometriosis, and infertility [31]. This is possibly because the up-regulation of HLA-DR can stimulate a host’s immune response, induce more immune cells to participate in the inflammatory response, and then accelerate the process of aortic atherosclerosis, ultimately leading to the occurrence of AAA. 

A previous study found that elevated HLA-DR+ immune cells can accelerate the development of AAA [25]. However, few studies have explored the molecular mechanisms underlying the impact of elevated HLA-DR on HLA-DR/NK cells in terms of the risk prevention of AAA. In our study, we conducted a mediation analysis with Mendelian randomization by incorporating a large number of metabolites and immune cells and found a total of 66 metabolites (including α-ketoglutarate, tryptophan, histidine, asparagine, leucine, and the AMP-to-threonine ratio) to have a causal relationship with the formation of an AAA. We also found that an increase in HLA-DR on HLA DR/NK cells would increase the AMP-to-threonine ratio, thereby affecting the progression of AAA. Previous studies have also found a decrease in ATP-related mitochondrial activity in abdominal aortic aneurysm tissue, which led to a decrease in ATP synthesis and an increase in AMP accumulation. This corresponds to the increased AMP-to-threonine ratio found in our study. The mitochondrial oxygen consumption rate coupled with ATP also significantly increased in mouse aortic smooth muscle cells treated with angiotensin I [32]. This suggests an increase in anaerobic metabolism within abdominal aortic aneurysm tissue, leading to the production of more AMP. Furthermore, some researchers have found that AMP activates protein kinase α2, inducing the formation of AAA in mice in vivo [33].

Threonine is an essential amino acid in the human body that can only be obtained through external sources. Hypoxia in abdominal aortic aneurysm tissue can also lead to a decrease in the content of threonine, which is in line with previous studies that have found a significant decrease in the content of leucine and isoleucine in abdominal aortic tissue. This supports our findings. 

Another interesting finding of this study is that a decrease in threonine content may be one of the risk factors for AAA. Threonine is an essential amino acid in the human body that can only be obtained through external sources. Although the relationship between threonine and the formation of an AAA has not been clearly reported in the literature, previous studies have found a significant decrease in leucine and isoleucine levels in AAA tissue, which may be associated with a poor prognosis [34]. We speculate that the decrease in the content of threonine and other essential amino acids may be associated with a decrease in ATP synthesis in abdominal aortic aneurysm tissue and a decrease in the ability of cells to actively uptake essential amino acids.

Our study has some limitations. First, the influence of horizontal pleiotropy on our study was not completely evaluated, even though multiple sensitivity analyses were conducted in this study. Second, by using public summarized data from other researchers rather than our own data, we could not conduct a subgroup analysis to further verify our results. Finally, although we used cross-validation to select the most suitable immune cells and metabolites, the low SNP frequency and the small number of AAA cases in the UK Biobank could have led to the loss of many instrumental variables and potential positive results. In the future, larger sample sizes from multiple databases are needed for cross-validation to obtain more robust results. Therefore, the conclusions of this study still need to be confirmed through prospective studies with larger sample sizes.

## 5. Conclusions

In this study, we found that the up-regulation of HLA-DR on HLA-DR/NK cells can increase the risk of AAA formation via improvements in the AMP-to-threonine ratio. Our results suggest that the AMP-to-threonine ratio might be adopted as a potential biomarker for the early prediction of AAA and for facilitating clinical decision-making.

## Figures and Tables

**Figure 1 biomedicines-12-01179-f001:**
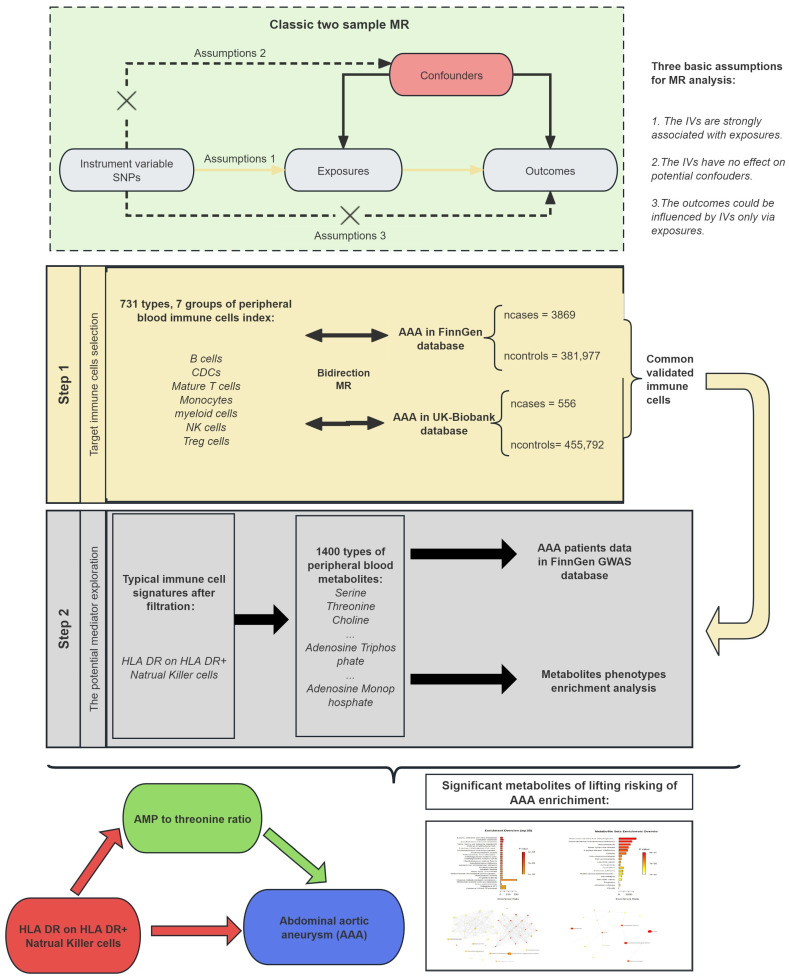
Diagram of work design and flow. Abbreviation: AAA: abdominal aortic aneurysm; MR: Mendelian randomized; GWAS: genome-wide association study; HLA: human leukocyte antigen; NK: natural killer; AMP: Adenosine 5’-monophosphate; SNP: single nucleotide polymorphisms; IVs: instrumental variables.

**Figure 2 biomedicines-12-01179-f002:**
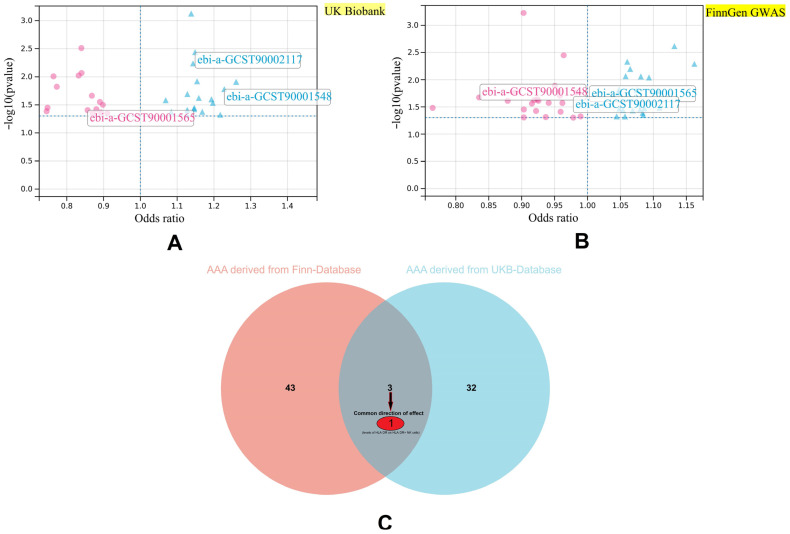
Two-sample MR filtration of key immune cell phenotypes with a causal effect with AAA in the two databases. (**A**) Volcano plots of the key immune cell phenotypes from the UK Biobank database separated by a significant causal effect with AAA. (**B**) Volcano plots of the key immune cell phenotypes from the FinnGen database separated by a significant causal effect with AAA. (**C**) A Venn diagram of commonality between the immune phenotypes. Abbreviations: AAA: abdominal aortic aneurysm; HLA: human leukocyte antigen; NK: natural killer; GWAS: genome-wide association study.

**Figure 3 biomedicines-12-01179-f003:**
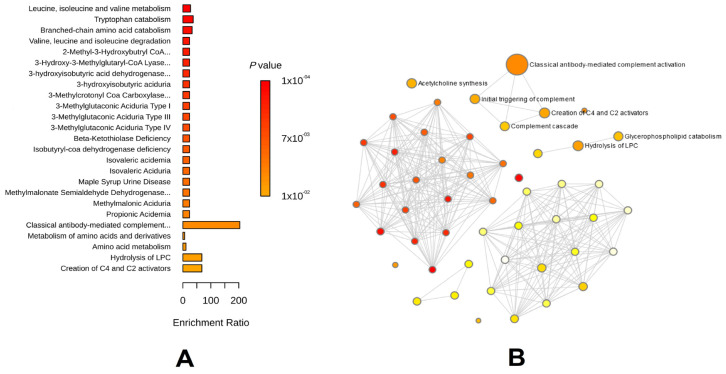
The enrichment results of peripheral blood metabolites phenotypes with casual relationships to AAA. (**A**) The metabolic enrichment plot of significantly protective metabolites phenotype to AAA. (**B**) The relationships network of metabolic functions calculating by significantly protective metabolites phenotype to AAA. (**C**) The metabolic enrichment plot of significantly risking metabolites phenotype to AAA. (**D**) The relationships network of metabolic functions calculating by significantly risking metabolites phenotype to AAA. (**E**) The common diseases enrichment plot of significantly risking metabolites phenotype to AAA. (**F**)The relationships network of common diseases enrichment calculating by significantly risking metabolites phenotype to AAA. Abbreviation: AAA: abdominal aortic aneurysm.

**Figure 4 biomedicines-12-01179-f004:**
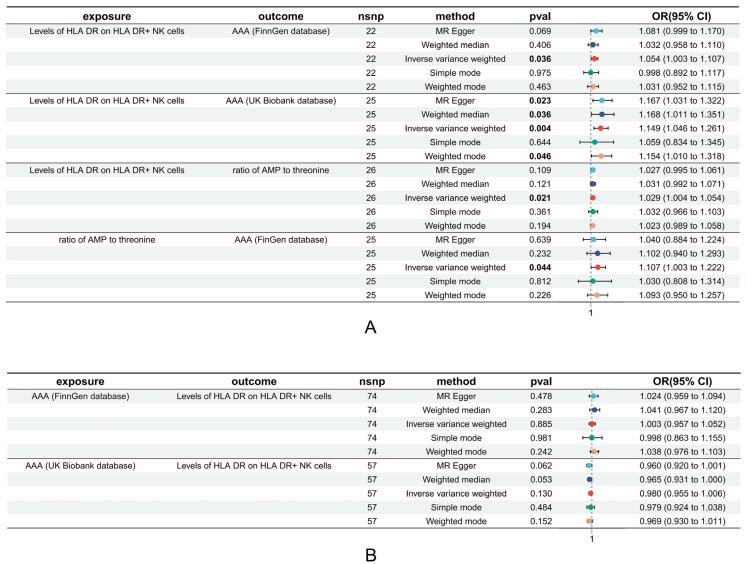
The forest plots of establishing mediation effect with 5 estimating methods included. (**A**) The forest plots of mediated effect estimated. (**B**) The forest plots of reverse two-sample MR analysis between the levels of HLA DR on HLA DR+ NK cells (outcome) and AAA of FinnGen and UK Biobank databases. ‘nsnp’ represents the number of SNPs included as instrumental variables; all statistically significant results with *p*-values less than 0.05 are highlighted in bold. In the forest plot, the statistical results of the five MR methods are represented by different colors: cyan, blue, red, green, and yellow, corresponding to the results of MR-Egger, Weighted Median, IVW, Simple Mode, and Weighted Mode methods, respectively. Abbreviation: AAA: abdominal aortic aneurysm; MR: Mendelian randomized; HLA: human leukocyte antigen; NK: natural killer; AMP: Adenosine 5’-monophosphate; OR: odds ratio; CI: confidence interval; nsnp: the number of SNPs.

## Data Availability

All data used in this study are publicly available genetic data. More details of this study can be found in the Appendix A.

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
