# Peer review of "Adenosine 5′-Monophosphate-to-Threonine Ratio Promotes Abdominal Aortic Aneurysms via Up-Regulation of HLA-DR on Natural Killer Cells: A Bidirectional Mendelian Randomized Analysis"

_biomedicines, 2024, doi:10.3390/biomedicines12061179_

Round 1

Reviewer 1 Report

Comments and Suggestions for Authors

Teng F. et al. reported that up-regulation of HLA-DR on HLA-DR+NK increases the risk of an AAA via improving the AMP-to-threonine ratio, thus providing a potential new biomarker in the prediction and treatment of AAA. This study addresses essential issues; however, the manuscript suffers from a few flows that require addressing to enhance clarity.

 Major issues:

1.    Clinical characteristics of the examined cohorts should be added to the manuscript.

2.    The presented data should be validated on the unrelated cohort (validation group).

3.    At least the specificity and sensitivity of HLA-DR as a potential biomarker in the prediction and treatment of AAA should be calculated.

4.    This observational study analyzed data deposited in public databases obtained from patients of European descent; therefore, I wonder whether the Authors plan to validate the presented results using clinical samples delivered from patients of Chinese descent?

Minor issues:

1.    The interval between sentences and reference number should be provided in the manuscript.

2.    The phrase “SNPSs” should be corrected to “SNPs”, page 10.

3.    Figures are unreadable and should be modified.

4.    A space in the Figure 3 legend should be added: (F) The relationship network of (…).

Comments on the Quality of English Language

A manuscript revision by a language editing service should be performed.

Author Response

Thank you for your timely and valuable feedback. We have once again commissioned an official agency to conduct a second round of polishing and will present the polished article to you. Thank you again for your feedback

  1. Response for“Clinical characteristics of the examined cohorts should be added to the manuscript.”:

Thank you for your valuable feedback. We also hope to obtain more detailed personalized GWAS information. However, due to personal privacy and genetic information security considerations, data publishers often do not provide such information to individuals, so the vast majority of researchers are unable to obtain it and therefore cannot create clinical information tables. However, there is a strict process for obtaining GWAS data at these summary levels, and these data providers have also made corresponding reports, such as the official websites of UK Biobank and FinnGen. At each stage of SNP effect evaluation, they also used a multiple linear regression model to consider the influence of the top ten factors, including age, gender, etc., in order to obtain a more accurate SNP effect. The most crucial aspect of MR implementation is the accuracy of SNP (i.e. instrumental variable) effect evaluation. Therefore, clinical information tables are not absolutely necessary in MR, but if they can be provided, it will definitely make the research more complete and scientific.

Thank you again for your valuable feedback. In the future, we will strive to apply for personalized GWAS large-scale research through our work unit to make up for this regret today.

  1. Response for“The presented data should be validated on the unrelated cohort (validation group)”:

Thank you for your correction. It is indeed necessary for any research validation. Our study was actually divided into three parts, but due to sample overlap issues, there were certain difficulties in cross validation. However, we have done our best to complete such validation. Our first part is the immunophenotype-AAA (FinnGen and UKB dual databases), which was mutually validated by the FinnGen and UKB dual databases to obtain the relationship between immunophenotypes and AAA. The validation of metabolites in the second part presents significant difficulties, as the two larger AAA related phenotype databases exist in FinnGen and UKB, but the sample size in UKB is too small compared to the frequency of SNPs (only over 500 cases, with 4000 cases in FinnGen). Therefore, the direction of validation is easy, but obtaining accurate statistical differences is difficult. In the future, a larger sample size database (at least 2000 cases should be larger to meet the problem of low SNP probability) is needed to avoid the loss of too many instrumental variables and the omission of positive results.

Thank you again for your feedback. This issue does indeed exist and your pointing out is very professional, but the cost of solving it is too high. However, we have fully discussed your opinions and believe that we should explain our results more cautiously. We have added this flaw to the discussion: Finally, although we use cross validation to select the most suitable immune cells and metabolites, the low SNP frequency and the small number of AAA cases in UK Biobank can lead to the loss of many instrumental variables and potential positive results. In the future, larger sample sizes of multiple databases are needed for cross validation to obtain more robust results.

Thank you again for your valuable feedback!

  1. Response for“At least the specificity and sensitivity of HLA-DR as a potential biomarker in the prediction and treatment of AAA should be calculated”:

Thank you for your correction. Indeed, as you mentioned, calculating sensitivity and specificity, as well as plotting ROC curves, are necessary for key molecular markers. We have also attempted to obtain specific individualized HLA-DR expression levels and individual level AAA incidence from multiple perspectives, but all were rejected. The main reason is that the data institution only provided data processing procedures and estimates of the effect sizes of tens of millions of SNPs per SNP, considering genetic privacy and security. So we may not be able to calculate sensitivity and specificity indicators based on HLA-DR expression levels, and can only estimate their mutual influence through MR. We will continue to work hard in this area and strengthen communication with data source institutions in other topics that extend from this topic. We will strive to not only evaluate potential causal relationships as you mentioned, but also conduct further exploration in specific numerical phenotype level studies. Thank you again for your valuable feedback.

  1. Response for“This observational study analyzed data deposited in public databases obtained from patients of European descent; therefore, I wonder whether the Authors plan to validate the presented results using clinical samples delivered from patients of Chinese descent?”:

Thank you for your feedback. In fact, we are also discussing the possibility of further extension of this paper. Firstly, as you have accurately pointed out, the issue of different genetic backgrounds in the population makes it difficult for the validation results in the Chinese population to be as ideal as in the European population. Secondly, validation requires a lengthy process and a significant amount of work, as the required validation is actually a causal relationship. Based on the above considerations, we actively contacted the basic research and experimental department of our hospital for further project discussions, and focused on discussing whether knockout/overexpression mice can complete the above validation. However, whether it is clinical validation or basic experimental validation, as you pointed out, this is a long-term work that requires careful design and consumes a lot of time and effort. We are currently in the argumentation stage.

Thank you again for providing us with professional and detailed valuable feedback amidst your busy schedule. Thank you very much!

  1. Response for“The interval between sentences and reference number should be provided in the manuscript”:

 Thank you for your reminder. We did indeed overlook this point, and we have reviewed the entire article and made revisions.

  1. Response for“The phrase “SNPSs” should be corrected to “SNPs”, page 10.”:

Thank you for your careful review. We have made the necessary changes regarding this error.

  1. Response for“Figures are unreadable and should be modified.”:

Thank you very much for your prompt reminder and recognition of our work. We have sought the original artist to increase the resolution, and we will upload the latest high-resolution images separately in a file. Thank you again for your reminder and we apologize for not providing high-quality illustrations in a timely manner.

  1. Response for“A space in the Figure 3 legend should be added: (F) The relationship network of (…).”:

Thank you for your reminder. We have made the necessary changes regarding this error. Thank you again.

Reviewer 2 Report

Comments and Suggestions for Authors

To be honest, I am a vascular surgeon, and mendelian randomization is not my field of expertise. I hope that the other reviewers will more extensively focus on the statistical and experimental methodology while I would like to give some recommendations from the clinical perspective. Overall, my impression is that the authors finished a tremendous and very specialized investigation considering this relatively new bioinformatics technology (although up to date, quering Pubmed with "mendelian randomization" lists more than 8,000 publication, the first one started only in 2019), but for the non-involved reader, it remains often a difficult task to follow the details of the M&M and the Results sections, and these should have been explained, at least in part, in a more comprehensive and logical way.

To start with the introduction section:

The first two references are somewhat misplaced, because the first one obviously is taken from a family medicine journal, and the second one has nothing to do with the sentence in which it was cited. There are a plenty of really more informative reviews on this topic in dedicated journals. The fourth reference is also not cited in the correct context, because it mentions the number of ~200,000 patients with diagnosis of AAA in the USA per year, not as an original finding, as may be expected, but obviously this number has been taken from another source and is therefore secondary literature. Citing literature should be done if possible from primary sources, with original publications and, if not historical, recent publication dates. 

"Abnormal activation of immune cells and degenerative changes in elastic and collagen fibers, are the main factors leading to AAA formation[4,10]."

It is known, that smoking, COPD, atherosclerosis and genetic predispositions are the main risk factors for the development of AAA and are worth being mentioned despite the focus of the manuscript on immunology. There are a plenty of epidemiological investigations, guidelines etc. handling with this. 

M&M section: the PLINK and R software packages, as freeware or public license tools, should be cited as suggested by the software authors. When a public database, such as the GWAS catalogue, is accessed, date and time should be mentioned. 

Results: Fig. 2 and 3 are of bad quality, the text within the figures is not readable. Figure 3 as a multipanel graphic is in general much too complex for one page and should be simplified or divided into several figures. The word "casual" appeared in lines 266 and 270, should it mean "causal"? The authors found upregulation of metabolites and HLA-DR levels to be associated with (prevalence or progress of) AAA, however upregulation seems to be a temporary physiological or pathological state of organ system homeostasis while AAA is a chronic disease. The authors should explain how their finding can be generalized to be a chronic predisposition. 

In the Discussion section, the authors summarize their results but plausible explanations are sometimes missing, such as for the fact that HLA-DR upregulation may be a risk factor for progression of AAA but a protector (against death?) in case of rupture. 

Dana Vucevic was mentioned as author of one of the cited papers. The author should pay attention that Dana is a female name. 

In general, hypothesized causal relationships between metabolites and AAA are described, but what are the mechanisms on the moleculare levels? The hypothesis that the ATP may be diminished in AAA is not very sound because ATP occurs exclusively within cells and is regulated very tightly as the end metabolite of all energy metabolism pathways. It is therefore an indicator of the amount of viable cells within the tissue. It seems that bioinformatics enlight only one aspect of biological causal relationships, it does not substitute clinical and experimental evidence, but adds new interesting insights.

Summarizing the aforementioned issues, some corrections should be done in the manuscript, after which the manuscript may be appropriate for publication.

Comments on the Quality of English Language

If the manuscript has not yet been edited by a native speaker, this is highly recommended because some parts of the text are difficult to read. 

Author Response

Thank you for providing professional advice and meticulous review amidst your busy schedule. We greatly appreciate your reminder of our research, which will help us improve the quality of our academic works and provide a better academic atmosphere. We have also submitted a request for further English polishing at the official English polishing agency and will present the revised manuscript to you. Thank you again for your feedback.

  1. Response for“The first two references are somewhat misplaced, because the first one obviously is taken from a family medicine journal, and the second one has nothing to do with the sentence in which it was cited. There are a plenty of really more informative reviews on this topic in dedicated journals. The fourth reference is also not cited in the correct context, because it mentions the number of ~200,000 patients with diagnosis of AAA in the USA per year, not as an original finding, as may be expected, but obviously this number has been taken from another source and is therefore secondary literature. Citing literature should be done if possible from primary sources, with original publications and, if not historical, recent publication dates. ”:

Thank you for your reminding. We reviewed the introduction section and the original citation manuscript and its related fields again. After review, we replaced the latest and more professional discussion article at Citation 2 to provide the latest definition of AAA, so as to ensure the accuracy and progressiveness of the discussion: Golledge J, Thanigaimani S, Powell JT, Tsao PS. Pathogenesis and management of abdominal aortic aneurysm. Eur Heart J. 2023;44(29):2682-2697. doi:10.1093/eurheartj/ehad386ï¼›The data cited in the fourth citation is from WHO's statistics on AAA in the United States. We have directly replaced the source of the original data with that citation: WHO . Cardiovascular diseases (CVDs). WHO; (2019). Available at: https://who.int/en/news-room/fact-sheets/detail/cardiovascular-diseases-(cvds) (Accessed July 05, 2020).

Thank you for your hard and careful correction!

  1. Response for“Abnormal activation of immune cells and degenerative changes in elastic and collagen fibers, are the main factors leading to AAA formation[4,10].

It is known, that smoking, COPD, atherosclerosis and genetic predispositions are the main risk factors for the development of AAA and are worth being mentioned despite the focus of the manuscript on immunology. There are a plenty of epidemiological investigations, guidelines etc. handling with this. ”:

Thank you for your rigorous correction. Previously, our discussion method made our perspective appear narrow. Now, we have changed this part of the discussion to: With the exploration of the pathogenesis of AAA this year, people found that in addition to smoking, COPD, atherosclerosis and genetic susceptibility are the main risk factors for the development of AAA, the immune system and metabonomics also play a crucial role in the occurrence and development of AAA [6-9]. The abnormal activation of immune cells and the degenerative changes of elastic fibers and collagen fibers are one of the main factors leading to the formation of AAA [4,10].

Thank you again for your criticism and correction!

  1. Response for“M&M section: the PLINK and R software packages, as freeware or public license tools, should be cited as suggested by the software authors. When a public database, such as the GWAS catalogue, is accessed, date and time should be mentioned. ”:

Thank you for your careful review. We apologize for any errors that may have occurred in our work. We have included the publication time of the data in the original text. As open source data, we were unable to find specific reference methods for R language and PLINK. After consideration, we have provided the corresponding version number: R language (version 4.3.1). In addition, based on your feedback, we have reviewed our article again and found that we are missing two references. Our suggested references for the UKB database and FinnGen database were omitted, and we have now added them: 21.Kurki, M.I., Karjalainen, J., Palta, P. et al. FinnGen provides genetic insights from a well-phenotyped isolated population. Nature 613, 508–518 (2023). https://doi.org/10.1038/s41586-022-05473-8

22.Bycroft C, Freeman C, Petkova D, Band G, Elliott LT, Sharp K, Motyer A,

Vukcevic D, Delaneau O, O’Connell J, et al. The UK Biobank resource with

deep phenotyping and genomic data. Nature. 2018;562(7726):203–9.

  1. Response for“Results: Fig. 2 and 3 are of bad quality, the text within the figures is not readable.”:

Thank you very much for your prompt reminder and recognition of our work. We have sought the original artist to increase the resolution, and we will upload the latest high-resolution images separately in a file. Thank you again for your reminder and we apologize for not providing high-quality illustrations in a timely manner.

  1. Response for“Figure 3 as a multipanel graphic is in general much too complex for one page and should be simplified or divided into several figures. The word "casual" appeared in lines 266 and 270, should it mean "causal"? The authors found upregulation of metabolites and HLA-DR levels to be associated with (prevalence or progress of) AAA, however upregulation seems to be a temporary physiological or pathological state of organ system homeostasis while AAA is a chronic disease. The authors should explain how their finding can be generalized to be a chronic predisposition. ”:

Thank you for your valuable feedback. We have split Figure 3 into Figure 3 and Figure 4 to make it less cumbersome.

Yes, due to the design of the research method using instrumental variable concept, MR can to some extent achieve statistical evaluation of causal relationships. But the effectiveness of its evaluation depends on the avoidance of pleiotropy and the exclusion of reverse causality. However, your reminder is quite correct, because although the design of MR can theoretically evaluate causality, in fact, no study can guarantee the 100% absence of pleiotropy. Although we have excluded the existence of these pleiotropy through multiple sensitivity and heterogeneity analyses, as well as reverse MR, we still cannot guarantee the absence of pleiotropy. Therefore, future clinical and laboratory experiments are necessary. So based on your feedback and careful consideration, we have chosen a more appropriate and conservative wording: This is the first time that a bidirectional two sample mediation analysis using MR has been conducted based on a large amount of public genetic data, aiming to investigate the possible causal relationships between multiple immune phenotypes, metabolites, and AAA after excluding the presence of pleiotropy as much as possible.

  Due to the fact that MR research is based on a large sample size (mainly SNP correlation p-values) in both exposed and outcome phenotypes, the expression level of HLA-DR phenotype is actually based on multiple period statistical results of more than 3700 subjects, and the instrumental variable actually represents the long-term expression results of HLA-DR. The outcome AAA indicator is also based on the statistical results of over 400000 people, who are also not in the same period. These instrumental variables SNPs, which can represent the expression level of HLA-DR phenotype, will continue to play a role in the outcome indicator (AAA) through HLA-DR. Since most SNPs do not change in individuals, they play a role from the determination of genetic material at the time of fertilization. Therefore, it is possible to attribute it to long-term effects.   

  Of course, your concerns also highlight a high level of professional expertise and domain understanding, which indeed highlights the key and necessity of future large-scale laboratory and biological research to confirm our research results. Thank you again for your valuable feedback, which has also made us more cautious in evaluating our research.

  1. Response for“In the Discussion section, the authors summarize their results but plausible explanations are sometimes missing, such as for the fact that HLA-DR upregulation may be a risk factor for progression of AAA but a protector (against death?) in case of rupture. ”:

Thank you for your professional advice in this regard. The standardized application of MR methods and the public access to GWAS databases are new changes in recent years, providing us with a new perspective to explore the previously inconvenient or costly disease molecular relationships from the perspective of MR. The causal relationships between molecules and diseases often found through MR lack comprehensive support from existing research. However, on the other hand, while excluding obvious pleiotropy, MR also provides new insights for future research.

 Specifically, in the case of AAA rupture, HLA-DR is a protective agent. This is a prospective observational study with a sample size of only 30 cases. The researchers found that the percentage of monocytes expressing HLA-DR in postoperative blood significantly decreased in patients who died after surgery compared to those who ultimately survived on days 3, 5, 7, 10, and 14. Therefore, we have cited this interesting point that goes against common sense. The reason why we haven't discussed too much here is that firstly, the research sample size here is small, and the randomness is high, so the results are not stable; Secondly, what we are studying is actually the relationship between HLA-DR and the pathogenicity of AAA, and what this article discusses here is actually the prognostic relationship between HLA-DR and AAA; Thirdly, the percentage of monocytes in HLA-DR used in this study is not entirely consistent with our HLA-DR expression level phenotype. However, your suggestions have also made us aware that there may be misunderstandings for readers here. After discussion, we have made revisions to this section. Thank you again for your professional and accurate feedback. The following is the revised section:

However, the upregulation of HLA-DR is related to the occurrence and development of AAA, but a prospective observational study also pointed out an interesting phenomenon that a decrease in the percentage of HLA-DR monocytes in patients who died after AAA surgery may indicate an increased risk of death, which may suggest that the expression of HLA-DR in different pathological stages and immune cells can lead to complex pathological and physiological outcomes.

  1. Response for“Dana Vucevic was mentioned as author of one of the cited papers. The author should pay attention that Dana is a female name. ”:

Thank you for your correction. We have made corrections in the paper, which is very embarrassing. Thank you for pointing it out in a timely manner, and we also thank Professor Dragana Vucevic for providing valuable academic insights and achievements.

Reviewer 3 Report

Comments and Suggestions for Authors

In this study, the authors conducted a two-sample, two-step mediation analysis using Mendelian randomization (MR) based on Genome-Wide Association Studies (GWAS) to investigate the causal relationships between blood immune cell signatures, metabolites, and AAA. These findings are interesting. However, the resolution of Figures 2 and 3 appears to be insufficient and the arrangement might need improvement. I encountered difficulty discerning the characters in these figures.

Comments on the Quality of English Language

The quality of English language is good enough for readers to understand the article.

Author Response

Thank you very much for your timely reminder and recognition of our work. We have sought the original artist to improve the resolution, and we will upload the latest high-resolution images separately in the file. Thank you again for your reminder. We apologize for not providing high-quality illustrations in a timely manner.

We have also submitted a request for further English polishing at the official English polishing agency and will present the revised manuscript to you. Thank you again for your feedback

Round 2

Reviewer 1 Report

Comments and Suggestions for Authors

The authors have satisfactorily addressed my concerns.

Minor comments:

1.         The abbreviation of COPD should be added to the 1. Introduction section, page 2.

Author Response

Thank you for pointing it out in time. We'll add  the abbreviation of COPD and reupload our manuscript. Thanks again for your feedback.